# Learning Optimal Combination Patterns for Lightweight Stereo Image Super-Resolution

## ABSTRACT

Stereo image super-resolution (stereoSR) strives to improve the quality of super-resolution by leveraging the auxiliary information provided by another perspective. Most approaches concentrate on refining module design, and stacking massive network blocks to extract and integrate information. Although there have been advancements, the memory and computation costs are increasing as well. To tackle this issue, we propose a lattice structure that autonomously learns the optimal combination patterns of network blocks, which enables the efficient and precise acquisition of feature representations, and ultimately achieves lightweight stereoSR. Specifically, we draw inspiration from the lattice phase equalizer and design lattice stereo NAFBlock (LSNB) to bridge pairs of NAFBlocks using re-weight block (RWBlock) through a coupled butterfly-style topological structures. RWBlock empowers LSNB with the capability to explore various combination patterns of pairwise NAFBlocks by adaptive re-weighting of feature. Moreover, we propose a lattice stereo attention module (LSAM) to search and transfer the most relevant features from another view. The resulting tightly interlinked architecture, named as LSSR, extensive experiments demonstrate that our method performs competitively to the state-of-the-art.

## KEYWORDS

Stereo image, super-resolution, lattice structure

## 1 INTRODUCTION

Stereo super resolution (stereo SR) images have garnered significant interest owing to their significant utility in 3D applications, such as depth estimation for autonomous vehicles [20] and computer-assisted surgery [21]. Nevertheless, in real-world scenarios, the generation of low-resolution (LR) image pairs is common due to acquisition limitations and the widespread use of cost-effective imaging systems. Consequently, the effective utilization of information from low resolution stereo images for stereo super resolution has emerged as a crucial undertaking.

Unlike single image SR (SISR), stereoSR performance relies not only on the intra-view information within the left and right images but also on the cross-view information between them. Hence, utilizing established SISR methods such as [15–17, 40, 41, 43] for independent reconstruction of the left and right images faces limitations in terms of performance due to the lack of cross-view information.

Recently, numerous deep learning approaches [3, 5, 13, 18, 19, 25, 36, 38] have been proposed for the stereoSR, yielding remarkable achievements. PASSRnet [33] first attempt to introduce a parallax-attention stereo super resolution network, which employs a global receptive field to effectively handle a diverse set of stereo images exhibiting substantial disparities. To enhance the effective integration of cross-view information, [1, 2, 34, 42] focus on the finely designing cross-attention module. Nevertheless, this results in an

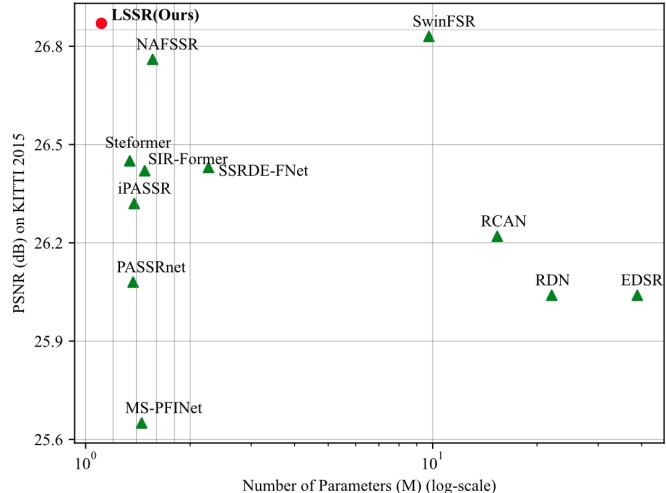

**Figure 1: The total number of parameters vs. PSNR of models for 4× stereo SR on KITTI 2015 [26] testset. Our LSSR achieve the SOFT performance with up to 89% of parameter reduction.**

increase in system complexity. To address this issue, [5] propose the stereo cross-attention modules (SCAM) between consecutive NAFBlocks [4], which blends the simplicity and effectiveness of NAFNet [4]. Leveraging the remarkable capabilities of the Transformer [30], [3, 9, 18, 38] devise Transformer-based models to reliably capture stereo correspondence and seamlessly integrate cross-view information for stereoSR. Despite the successive breakthroughs achieved by the aforementioned methods, they give rise to increased memory and computational demands, mainly as a consequence of refining modules and merely stacking multiple network blocks.

Based on the information presented, a natural question that comes to mind is whether it is feasible to design a learnable combination pattern instead of merely stacking massive network blocks? To achieve this goal, we design a lattice stereo super resolution architecture, named as LSSR. Specifically, we utilize NAFBlock [4] as the base block and concentrate on designing a novel combination pattern. Drawing lessons from the butterfly-style topological structures of the lattice phase equaliser [14, 24], we propose the lattice stereo NAFBlock (LSNB). LSNB is a versatile block that connects pairs of NAFBlocks using the re-weight block (RWBlock) within coupled butterfly-style topological structures. RWBlock adaptively reweighting input features empowers LSNB to explore the various linear combination patterns between pairwise NAFBlocks rather than merely stacking numerous network blocks, resulting in lightweight stereoSR through the utilization of the learned optimal combination pattern. What's more, we propose a lattice stereo attention

module (LSAM) to enhance the effective information exchange between two views. It initially calculates bidirectional cross-attention from both left to right and right to left views and subsequently merges the mutually influenced cross-view features with intra-view features. Figure. 1 shows our LSSR achieves state-of-the-art performance while also maintaining a lightweight parameter.

The main contributions of this work are:

(1) We introduce a lightweight lattice stereo super resolution approach named LSSR, which learns the optimal combination patterns of network blocks, effectively extracts intra-view features, and seamlessly integrates cross-view features. Extensive experiments are conducted to demonstrate the effectiveness and efficiency of our proposed LSSR.

(2) We propose a novel lattice stereo NAFBlock (LSNB) that leverages interconnected butterfly-style topological structures. LSNB inherits the simplicity and effectiveness of NAF-Block, while enhancing the model's representation by adaptively adjusting various combinations of pairwise NAFBlocks.

(3) We introduce a re-weight block (RWBlock) within LSNB, which leverages the attention mechanism to obtain re-weighted features, facilitating the flexible exploration of various combination patterns.

(4) We design a lattice stereo attention module (LSAM) to enhance the effectiveness of information exchange between intra-view and cross-view features.

## 2 RELATED WORKS

### 2.1 Single Image Super-resolution

Single image super-resolution (SISR) is a longstanding problem that has been under investigation for decades [15, 27, 37, 43]. Its objective is to generate high-resolution images solely based on intra-view information derived from low-resolution counterparts. SRCNN [7] makes the initial foray into applying deep learning to SISR by employing a three-layer convolutional neural network for the SR task. In order to enhance the model's representation capabilities, increasingly intricate models are being devised. VDSR [11] and EDSR [17] amplify the model's depth and width while implementing skip connections for residual information learning, thereby preventing gradient collapse. CBAM [35] utilizes channel and spatial attention blocks to extract the contextual relations as highly effective tools for addressing SISR. Transformers [16, 30, 39] have been applied to SISR, demonstrating commendable performance by effectively capturing non-local information. SwinIR [16] proposes an image restoration method based on the Swin Transformer [22] and achieves state-of-the-art performance on SISR.

### 2.2 Stereo Image Super-resolution

Stereo image super-resolution (stereoSR) is dedicated to restoring high-resolution details in both the left and right views of stereo image pairs by leveraging cross-view information [31, 32]. StereoSR [10] learns a parallax prior by jointly training two cascaded sub-networks for luminance and chrominance, integrating cross-view information by concatenating the left image with right images with predefined shifts. However, it has limitations in handling scenes with significant disparity variations due to fixed shift intervals. To handle this issue, PASSRnet [33] introduces a parallax attention

module to acquire stereo correspondence. iPASSR [34] integrates a symmetric bi-directional parallax attention module (biPAM) and an inline occlusion handling scheme to effectively utilize symmetry cues. And NAFSSR [5] yields impressive results by interposing simple cross-view attention modules (SCAMs) between consecutive NAFBlocks [4]. Transformer-based models [3, 18, 38] are now being utilized in stereoSR because of their capacity to capture long-range dependencies within images. SIR-Former [38] is the first to introduce transformers into stereo image super-resolution, utilizing a cross-attention module to learn epipolar line relationships and a transformer-based fusion module for accurate cross-view feature integration. Then, SwinFSR [3] introduces an extended StereoSR method, building upon the SwinIR [16] foundation and leveraging frequency domain knowledge acquired through fast fourier convolution [29]. Futhermore, Steformer [18] leverages Transformer's self-attention for capturing both cross-view and intra-view information in stereo image, ensuring dependable stereo correspondence and cross-view integration. While all of the mentioned models have shown significant performance enhancements, they primarily rely on stacking numerous blocks and do not delve into the combination patterns among blocks.

In this paper, we introduce a novel lattice stereo NAFBlock (LSNB) that leverages coupled butterfly-style topological structures. This design allows to learn the optimal combination patterns between pairwise NAFBlocks. Additionally, we have designed a lattice stereo attention module (LSAM) to effectively seamlessly integrate cross-view information for stereoSR.

## 3 METHOD

Our primary goal is to investigate a adaptively regulated combination pattern instead of merely stacking massive network blocks in the model. With this goal in mind, we present a lattice stereo super resolution network (LSSR) shown in Figure. 2. LSSR employs two weight-sharing branches constructed with lattice stereo NAFBlock (LSNB) to separately extract intra-view features from the left and right images. Additionally, lattice stereo attention modules (LSAMs) are incorporated to combine cross-view features.

**Overall Pipeline.** Given a pair of low resolution stereo images $I_L \in \mathbb{R}^{H \times W \times 3}$ (left view) and $I_R \in \mathbb{R}^{H \times W \times 3}$ (right view), LSSR first applies a $3 \times 3$ convolutional layer to extract shallow feature maps $F_L^S \in \mathbb{R}^{H \times W \times C}$, $F_R^S \in \mathbb{R}^{H \times W \times C}$ ($H, W, C$ are the feature map height, width, and channel number, respectively). Next, these shallow features $F_L^S$, $F_R^S$ pass through $N$ LSNBs to achieve deep intra-view feature extraction. To interact with cross-view information, we incorporate LSAM following each LSNB. After the aforementioned process, we acquire deep features denoted as $F_L^D$ and $F_R^D$, encompassing both intra-view and cross-view information. Furthermore, we apply a $3 \times 3$ convolution layer followed by a pixel shuffle layer to upsample the deep feature by a scale factor of $s$, and generate $R_L \in R^{H \times W \times 3}$, $R_R \in R^{H \times W \times 3}$. Noted, to alleviate the burden of feature extraction, the $R_L \in R^{H \times W \times 3}$ and $R_R \in R^{H \times W \times 3}$ are the difference between the bilinearly upsampled low-resolution image and the high-resolution ground truth. Thus, the $R_L + I_L$, $R_R + I_R$ are the high-resolution images of the left and right views, respectively.

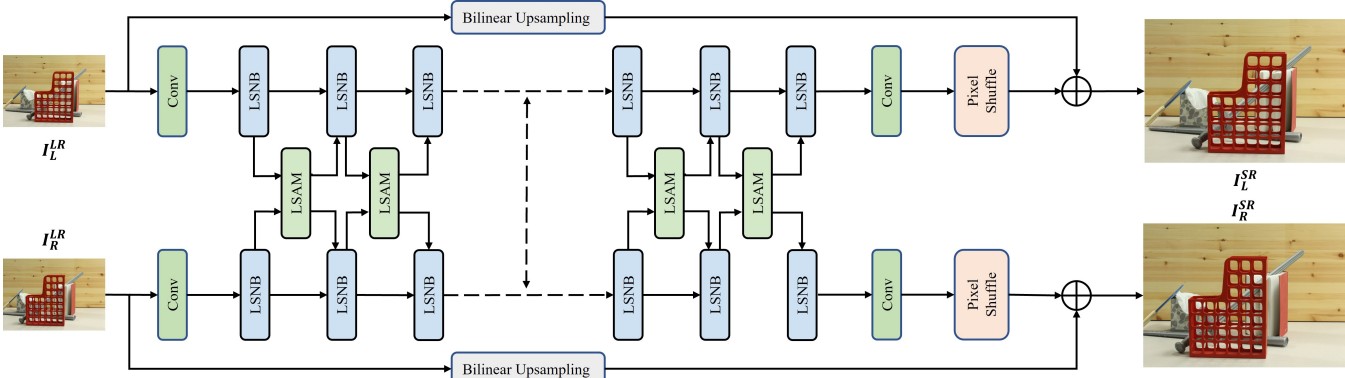

**Figure 2: The overall architecture of LSSR with two key conponents: (1) lattice stereo NAFBlock (LSNB) (illustrated in Figure. 3(a)) and lattice stereo attention module (LSAM) (depicted in Figure. 5)**

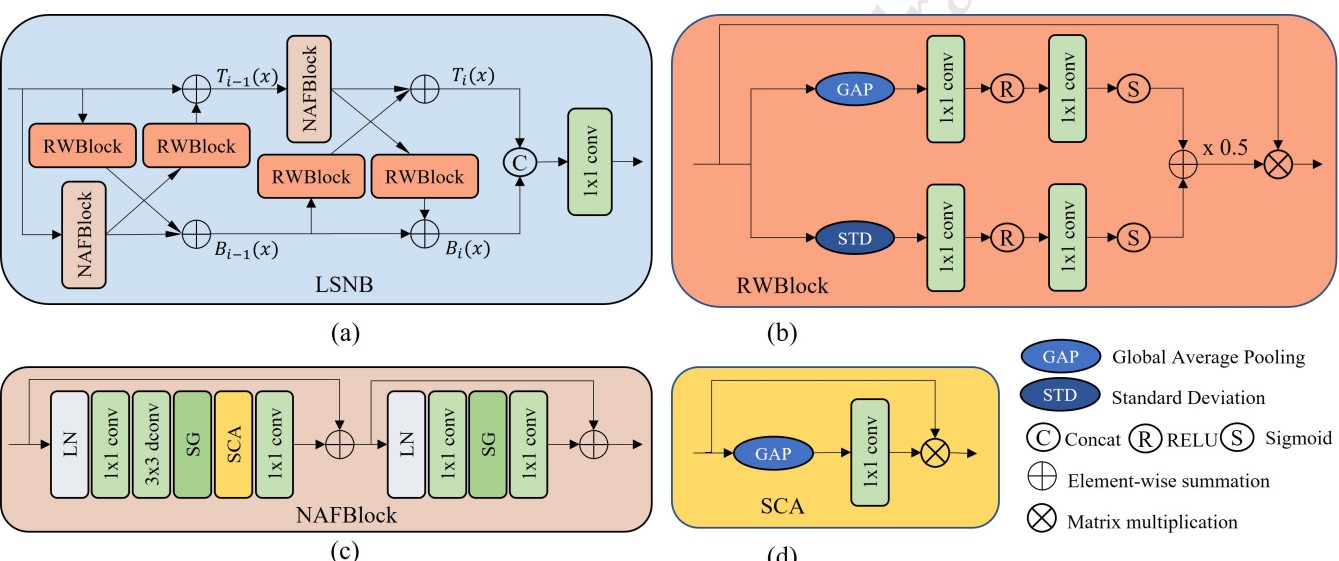

**Figure 3: (a) Lattice stereo NAFBlock (LSNB) that provides various linear combination patterns of pairwise NAFBlocks [4]. (b) Re-weight block(RWBlock) that re-weight the features based on the attention mechanism. (c) The architecture of nonlinear activation free block (NAFBlock) [4]. (d) Simplified channel attention (SCA).**

## 3.1 Lattice Stereo NAFBlock

In pursuit of comprehensive and precise features, [3, 18, 34, 38] concentrates on refining module design and stacking massive network blocks, despite the resource-intensive nature of this approach. To address these limitations, inspired by the butterfly-style topological structures [24], we design a lattice stereo NAFBlock (LSNB) as the basic building block from the perspective of network block combination patterns. As shown in Figure. 3(a), we use the nonlinear activation free block (NAFBlock) [4] as the base block and employ the re-weight block (RWBlock) adaptively reweighting input features empowers LSNB to learn the optimal combination pattern between pairwise NAFBlocks. To be specific, given a input features

$\mathbf{x}$, the first combination can be defined as :

$$T_{i-1}(\mathbf{x}) = RW_2(NAF_1(\mathbf{x})) + \mathbf{x}$$
$$B_{i-1}(\mathbf{x}) = NAF_1(\mathbf{x}) + RW_1(\mathbf{x}) \tag{1}$$

where $NAF_i(\cdot)$ denotes the $i$-th NAFBlock, and $RW_j(\cdot)$ represents the $j$-th RWBlock, which are described below. Then, $T_{i-1}(\mathbf{x})$, $B_{i-1}(\mathbf{x})$ are used as input and a second combination is performed as:

$$T_i(\mathbf{x}) = RW_3(B_{i-1}(\mathbf{x})) + NAF_2(T_{i-1}(\mathbf{x}))$$
$$B_i(\mathbf{x}) = RW_4(NAF_2(T_{i-1}(\mathbf{x}))) + B_{i-1}(\mathbf{x}) \tag{2}$$

Afterwards, the results of the second combination $T_i(\mathbf{x})$ and $B_i(\mathbf{x})$ are combined in the channel dimension and subsequently subjected to a $1 \times 1$ convolution to transform them to their original dimensions.

Based on the formula provided above, it becomes evident that the output feature result following RWBlock re-weighting has an influence on the composition of both $T_i(\mathbf{x})$ and $B_i(\mathbf{x})$. RWBlock empowers LSNB with the capability to explore various combination patterns of pairwise NAFBlocks. Next, we will provide a detailed explanation of several candidate structures for LSNB following the application of different re-weight results.

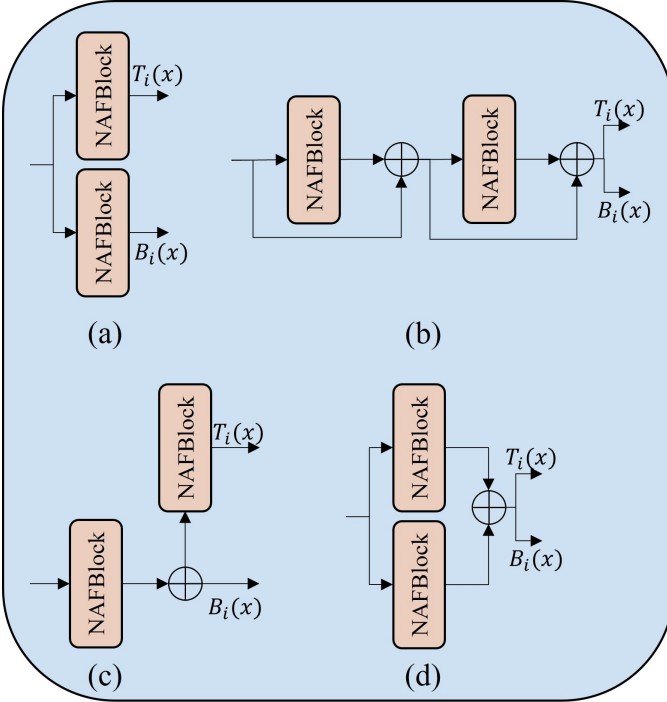

**Figure 4: Candidate structure examples of LSNB that emerge after applying various re-weighting results.**

In the following cases, the bold "**0**" denote vectors with all elements being 0:

(1) If $RW_1(\mathbf{x}) = RW_2(\mathbf{x}) = RW_3(\mathbf{x}) = RW_4(\mathbf{x}) = \mathbf{0}$, then Eq. 2 can be reformulated as follows:
$$T_i(\mathbf{x}) = NAF_2(\mathbf{x})$$
$$B_i(\mathbf{x}) = NAF_1(\mathbf{x}) \tag{3}$$

In this case, $T_i(\mathbf{x}) \neq B_i(\mathbf{x})$, the structure of LSNB is simplified as two concurrent NAFBlocks as shown in Figure. 4(a).

(2) If $RW_1(\mathbf{x}) = RW_2(\mathbf{x}) = RW_3(\mathbf{x}) = RW_4(\mathbf{x}) = \mathbf{x}$, then Eq. 2 can be reformulated as follows:
$$T_i(\mathbf{x}) = NAF_1(\mathbf{x}) + x + NAF_2(NAF_1(\mathbf{x}) + \mathbf{x})$$
$$B_i(\mathbf{x}) = NAF_1(\mathbf{x}) + x + NAF_2(NAF_1(\mathbf{x}) + \mathbf{x}) \tag{4}$$

In this case, $T_i(\mathbf{x}) = B_i(\mathbf{x})$, the structure of LSNB is simplified as cascade connections of pair-wise NAFBlocks as shown in Figure. 4(b).

(3) If $RW_1(\mathbf{x}) = RW_2(\mathbf{x}) = \mathbf{x}$, $RW_3(\mathbf{x}) = RW_4(\mathbf{x}) = \mathbf{0}$, then Eq. 2 can be reformulated as follows:
$$T_i(\mathbf{x}) = NAF_2(\mathbf{x} + NAF_1(\mathbf{x}))$$
$$B_i(\mathbf{x}) = NAF_1(\mathbf{x}) + \mathbf{x} \tag{5}$$

In this case, $T_i(\mathbf{x}) \neq B_i(\mathbf{x})$, as shown in Figure. 4(c), the structure of LSNB is simplified as a NAFBlock following a NAFBlock, the final result is the output of each block like [4, 5].

(4) If $RW_1(\mathbf{x}) = RW_2(\mathbf{x}) = \mathbf{0}$, $RW_3(\mathbf{x}) = RW_4(\mathbf{x}) = \mathbf{x}$, then Eq. 2 can be reformulated as follows:
$$T_i(\mathbf{x}) = NAF_1(\mathbf{x}) + NAF_2(\mathbf{x})$$
$$B_i(\mathbf{x}) = NAF_1(\mathbf{x}) + NAF_2(\mathbf{x}) \tag{6}$$

In this case, $T_i(\mathbf{x}) = B_i(\mathbf{x})$, the structure of LSNB is simplified as two parallel NAFBlocks as shown in Figure. 4(d).

Beyond the special cases mentioned earlier, the RWBlock can also dynamically re-weight features using the attention mechanism, leading to the inclusion of numerous other potential candidate structures within LSNB. To put it differently, the varied combination patterns provided by LSNB allow us to learn the optimal combination pattern for for designing lightweight models, avoiding the approach of simply stacking a large number of modules.

*3.1.1 Re-weight Block.* The Re-weight Block (RWBlock) serves as a crucial connection bridge in LSNB (see Figure. 3(a)) playing a key role. To flexibly adapt the combination patterns, instead of exhaustively searching all possible combinations, we utilize the attention mechanism to learn the re-weighted features.

As Figure. 3(b) shows, RWBlock consists of two branches: in the upper branch, we calculate the mean value of input features, while in the lower branch, we compute the standard deviation of input features. Following this, each branch are separately processed by two $1 \times 1$ convolution layers. Each of these convolution layers is succeeded by a RELU activation layer and a Sigmoid activation layer. Subsequently, the output from the two branches are averaged to obtain the reweight coefficients. Finally, these reweight coefficients are multiplied by the input features to obtain the re-weighted features. In this way, given an input feature $X_f$, the entire feature re-weight procedure of the developed RWBlock is formulated as:

$$X_{uc} = Sigmoid(f_{1\times1}^c(RELU(f_{1\times1}^c(GAP(X_f)))))$$
$$X_{lc} = Sigmoid(f_{1\times1}^c(RELU(f_{1\times1}^c(STD(X_f)))))$$
$$X_{cc} = (X_{uc} + X_{lc}) \times 0.5 \tag{7}$$
$$X_{rf} = X_f \otimes X_{cc}$$

where $f_{1\times1}^c$ represent $1 \times 1$ convolution, GAP is the global average pooling, STD is the standard deviation, $X_{cc}$ is the learned reweight coefficients, and $X_{rf}$ is the re-weighted features.

*3.1.2 NAFBlock.* As previously mentioned in our introduction, we use the nonlinear activation free block (NAFBlock) [4] as the base block in our LSNB. Fig. 3(c) illustrates the process of obtaining an output $NAF(X)$ from an input $X$ using Layer Normalization (LN), Convolution, Simple Gate (SG), and Simplified Channel Attention (SCA). Express as follows:

$$X_1 = X + f_{1\times1}^c(SCA(SG(f_{3\times3}^{dwc}(f_{1\times1}^c(LN(X))))))$$
$$SG = X_{f1} \cdot X_{f2}$$
$$SCA = X_{f3} \cdot f_{1\times1}^c(GAP(X_{f3})) \tag{8}$$
$$NAF(X) = X_1 + f_{1\times1}^c(SG(f_{1\times1}^c(LN(X_1))))$$

where $f_{3\times3}^{dwc}$ is the $3 \times 3$ depth-wise convolution, and $X_{f1}, X_{f2} \in \mathbb{R}^{H \times W \times \frac{C}{2}}$ are obtained by dividing $X_{f0}$ into channel dimensions. For a more intuitive presentation, we show $SCA(\cdot)$ in Fig. 3(d).

## 3.2 Lattice Stereo Attention Module

We reexamine all the prior cross-attention modules [3, 5, 34], where we compute the dot products between the query $Q \in \mathbb{R}^{H \times W \times C}$ projected by the source intra-view feature (e.g., left-view), and the key,value $K, V \in \mathbb{R}^{H \times W \times C}$ projected using the target intra-view feature (e.g., right-view). Followed by applying a softmax function to derive the weights assigned to the values:

$$Attention(Q, K, V) = softmax(\frac{QK^T}{\beta})V \qquad (9)$$

where $\beta$ is a learning scaling parameter used to adjust the magnitude of the dot product of $Q$ and $K$ prior to the application of the softmax function defined by $\beta = \sqrt{C}$.

Inspired by [3, 24], we propose a lattice stereo attention module (LSAM) to effectively capture reliable stereo correspondences and seamlessly integrate cross-view information for stereoSR, as shown in Figure. 5. Specifically, given a pair of intra-view features $X_L, X_R \in \mathbb{R}^{H \times W \times C}$, we begin by applying layer normalization, and subsequently derive the feature $\hat{X}_L$ and $\hat{X}_R$ using LSNB in conjunction with a $1 \times 1$ convolution layer. Then, we follow [3, 34] to feed $\hat{X}_L$ and $\hat{X}_R$ to a whitening layer, obtaining normalized features for establishing disentangled pairwise parallax attention, as defined by the following two equations:

$$X_L'(h, w, c) = \hat{X}_L(h, w, c) - \frac{1}{W}\sum_{i=1}^{W}\hat{X}_L(h, i, c)$$

$$X_R'(h, w, c) = \hat{X}_R(h, w, c) - \frac{1}{W}\sum_{i=1}^{W}\hat{X}_R(h, i, c) \qquad (10)$$

Noted that, $Q_L = X_L'$, $K_L = X_R'$ and $Q_R = X_R'$, $K_R = X_L'$. Next, we generate the value matrix $V_L$ and $V_R$ by using a $1 \times 1$ convolution layer, respectively. Subsequently, we compute bidirectional cross-attention between the left and right views as follows:

$$F_{R \to L} = Attention(Q_R, K_R, V_R)$$

$$F_{L \to R} = Attention(Q_L, K_L, V_L) \qquad (11)$$

Finally, the interacted cross-view information $F_{R \to L}$, $F_{L \to R}$ and intra-view information $X_L$, $X_R$ are fused by element-wise addition:

$$F_L = X_L + \lambda_L F_{R \to L}$$

$$F_R = X_R + \lambda_R F_{L \to R} \qquad (12)$$

where $\lambda_L$ and $\lambda_R$ are channel-wise scale parameters that are trainable and initialized with zeros to aid in stabilizing training.

## 3.3 Training Strategies

**Data augmentation.** In stereoSR tasks, it's common to apply random horizontal and vertical flips for dataset diversity. Furthermore, to enhance data utilization, we employ channel shuffling, randomly rearranging the RGB channels for color augmentation.

**Loss.** Following [5], we only use the pixel-wise L1 distance between the SR and ground-truth stereo images:

$$L = \|I_L^{SR} - I_L^{HR}\|_1 + \|I_R^{SR} - I_R^{HR}\|_1 \qquad (13)$$

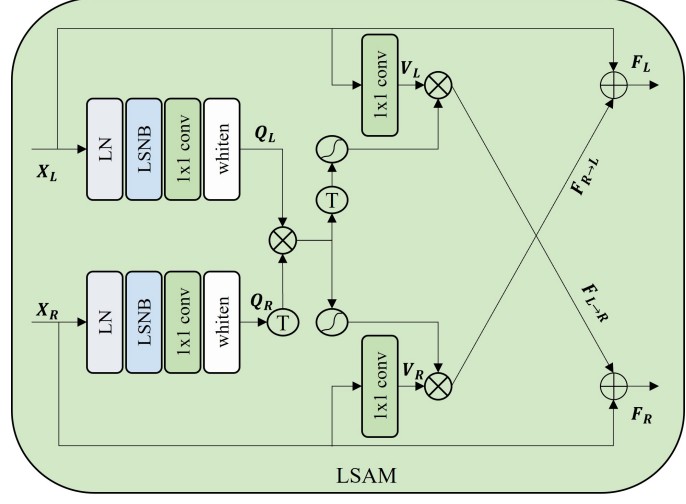

**Figure 5: Lattice stereo attention module (LSAM) that enables the interaction of cross-view information with intra-view information.**

where $I_L^{SR}$ and $I_R^{SR}$ represent the super-resolved left and right images, and $I_L^{HR}$ and $I_R^{HR}$ represent their ground-truth high-resolution images.

## 4 EXPERIMENTS

### 4.1 Implementation Details

**Datasets.** To train the proposed network, we utilize training data identical to that of [5, 34]. Specifically, we collected 800 images from the Flickr1024 dataset [33] and 60 images from the Middlebury dataset [28] for our training dataset. Since the images from the Middlebury dataset have a much higher spatial resolution than others, we perform bicubic downsampling with a scale factor of 2 to generate HR images from them. To generate LR images, we apply bicubic downsampling to HR images using specific scaling factors (i.e., 2× and 4×). The resulting LR images were then cropped into $30 \times 90$ patches with a stride of 20, and their HR counterparts were cropped accordingly. Finally, we obtained a total of 49,020 patches for 4× SR training and 298,143 patches for 2× SR training.

**Evaluation details.** To assess the performance of the proposed network, we employed a test dataset consisting of 112 images from the Flickr1024 dataset [33], 5 images from the Middlebury dataset [28], 20 images from the KITTI 2012 dataset [8], and 20 images from the KITTI 2015 dataset [26]. To achieve fair comparison with [5, 33, 34], we report Peak signal-to-noise ratio (PSNR) and structural similarity (SSIM) on the left views while cropping their left boundaries (64 pixels), and the average scores on stereo image pairs (i.e., $(Left + Right)/2$) without any boundary cropping.

**Training details.** We train our LSSR using the Adam [12] optimizer ($\beta_1 = 0.9, \beta_2 = 0.9$) and the batch size was fixed at 32 for $1 \times 10^5$ iterations. We initiated training with a learning rate of

**Table 1: Quantitative results achieved by various methods on the *KITTI*2012 [8], *KITTI*2015 [26], *Middlebury* [28] and *Flickr*1024 [33] datasets. The number of network parameters denoted as #P. The reported values include PSNR/SSIM results for both the left images (i.e., *Left*) and a pair of stereo images (i.e., (*Left* + *Right*)/2. The best results are highlighted in bold.**

| Method | Scale | #P | *Left* | | | (*Left* + *Right*)/2 | | | |
|---|---|---|---|---|---|---|---|---|---|
| | | | KITTI2012 | KITTI2015 | Middlebury | KITTI2012 | KITTI2015 | Middlebury | Flicker1024 |
| EDSR [17] | x2 | 38.6M | 30.83/0.9199 | 29.94/0.9231 | 34.84/0.9489 | 30.96/0.9228 | 30.73/0.9335 | 34.95/0.9492 | 28.66/0.9087 |
| RDN [41] | ×2 | 22.0M | 30.81/0.9197 | 29.91/0.9224 | 34.85/0.9488 | 30.94/0.9227 | 30.70/0.9330 | 34.94/0.9491 | 28.64/0.9084 |
| RCAN [40] | ×2 | 15.3M | 30.88/0.9202 | 29.97/0.9231 | 34.80/0.9482 | 31.02/0.9232 | 30.77/0.9336 | 34.90/0.9486 | 28.63/0.9082 |
| SwinIR [16] | x2 | 1.32M | 30.89/0.9206 | 29.98/0.9237 | 34.69/0.9475 | 31.02/0.9235 | 30.77/0.9341 | 34.80/0.9478 | 28.67/0.9091 |
| PASSRnet [33] | ×2 | 1.37M | 30.68/0.9159 | 29.81/0.9191 | 34.13/0.9421 | 30.81/0.9190 | 30.60/0.9300 | 34.23/0.9422 | 28.38/0.9038 |
| iPASSR [34] | x2 | 1.38M | 30.97/0.9210 | 30.01/0.9234 | 34.41/0.9454 | 31.11/0.9240 | 30.81/0.9340 | 34.51/0.9454 | 28.60/0.9097 |
| SSRDE-FNet [6] | ×2 | 2.10M | 31.08/0.9224 | 30.10/0.9245 | 35.02/0.9508 | 31.23/0.9254 | 30.90/0.9352 | 35.09/0.9511 | 28.85/0.9132 |
| SIR-Former [38] | ×2 | 1.37M | 31.02/0.9217 | 30.11/0.9246 | 34.87/0.9490 | 31.16/0.9247 | 30.93/0.9355 | 34.95/0.9495 | 28.69/0.9103 |
| SCVSCA [1] | x2 | 2.46M | 30.98/0.9129 | 30.04/0.9161 | 34.96/0.9436 | 31.12/0.9162 | 30.83/0.9273 | 35.02/0.9434 | 28.87/0.9035 |
| Steformer [18] | x2 | 1.29M | 31.16/0.9236 | 30.27/0.9271 | 35.15/0.9512 | 31.29/0.9263 | 31.07/0.9371 | 35.23/0.9511 | 28.97/0.9141 |
| NAFSSR [5] | x2 | 1.54M | 31.23/0.9236 | **30.28**/0.9266 | 35.23/0.9515 | 31.38/0.9266 | 31.08/0.9367 | 35.30/0.9514 | **29.19**/0.9160 |
| **LSSR (Ours)** | x2 | 1.09M | **31.26/0.9245** | **30.28/0.9273** | **35.33/0.9530** | **31.40/0.9275** | **31.09/0.9373** | **35.39/0.9530** | 29.15/**0.9171** |
| EDSR [17] | x4 | 38.9M | 26.26/0.7954 | 25.38/0.7811 | 29.15/0.8383 | 26.35/0.8015 | 26.04/0.8039 | 29.23/0.8397 | 23.46/0.7285 |
| RDN [41] | ×4 | 22.0M | 26.23/0.7952 | 25.37/0.7813 | 29.15/0.8387 | 26.32/0.8014 | 26.04/0.8043 | 29.27/0.8404 | 23.47/0.7295 |
| RCAN [40] | ×4 | 15.4M | 26.36/0.7968 | 25.53/0.7836 | 29.20/0.8381 | 26.44/0.8029 | 26.22/0.8068 | 29.30/0.8397 | 23.48/0.7286 |
| SwinIR [16] | x4 | 1.35M | 26.43/0.7996 | 25.60/0.7868 | 29.16/0.8379 | 26.52/0.8058 | 26.29/0.8098 | 29.25/0.8385 | 23.53/0.7322 |
| PASSRnet [33] | ×4 | 1.42M | 26.26/0.7919 | 25.41/0.7772 | 28.61/0.8232 | 26.34/0.7981 | 26.08/0.8002 | 28.72/0.8236 | 23.31/0.7195 |
| iPASSR [34] | x4 | 1.42M | 26.47/0.7993 | 25.61/0.7850 | 29.07/0.8363 | 26.56/0.8053 | 26.32/0.8084 | 29.16/0.8367 | 23.44/0.7287 |
| SSRDE-FNet [6] | ×4 | 2.24M | 26.61/0.8028 | 25.74/0.7884 | 29.29/0.8407 | 26.70/0.8082 | 26.43/0.8118 | 29.38/0.8411 | 23.59/0.7352 |
| SIR-Former [38] | ×4 | 1.48M | 26.53/0.7998 | 25.75/0.7882 | 29.23/0.8396 | 26.68/0.8077 | 26.42/0.8098 | 29.32/0.8407 | 23.52/0.7305 |
| SCVSCA [1] | x4 | 2.46M | 26.58/0.7864 | 25.73/0.7736 | 29.30/0.8286 | 26.68/0.7932 | 26.44/0.7974 | 29.40/0.8285 | 23.64/0.7186 |
| Steformer [18] | x4 | 1.34M | 26.61/0.8037 | 25.74/0.7906 | 29.29/0.8424 | 26.70/0.8098 | 26.45/0.8134 | 29.38/0.8425 | 23.58/0.7376 |
| NAFSSR [5] | x4 | 1.56M | 26.84/0.8086 | 26.03/0.7978 | 29.62/0.8482 | 26.93/0.8145 | 26.76/0.8203 | 29.72/0.8490 | **23.88/0.7468** |
| **LSSR (Ours)** | x4 | 1.11M | **26.93/0.8097** | **26.12/0.7997** | **29.86**/0.8489 | **27.02/0.8157** | **26.87/0.8226** | **29.92/0.8492** | 23.87/0.7432 |

$3 \times 10^{-3}$ gradually reduced it to $1 \times 10^{-7}$ with the cosine annealing [23].

## 4.2 Comparison with the State-of-the-Arts

We compare our LSSR with existing SR methods, encompassing both SISR methods [16, 17, 40, 41] and stereoSR methods [1, 5, 6, 18, 33, 34, 38]. Note that, for a fair comparison, we retrained all of these methods using our training dataset.

**Quantitative results.** As the quantitative results shown in Table. 1, LSSR achieves considerable results on all datasets [8, 26, 28, 33] and upsampling factors (×2, ×4) with a lower number of parameters. More specifically, in the case of 4× stereo SR, with 29% fewer network parameters, our LSSR outperforms the previous state-of-the-art model NAFSSR [5] by 0.09 dB, 0.11 dB, 0.20 dB on KITTI 2012 [8], KITTI 2015 [26] and Middlebury datasets [28], which demonstrates the effectiveness of the proposed LSSR.

**Lightweight and Efficiency.** We presented a visualization of the trade-off results between the total number of parameters and PSNR on the KITTI 2015 dataset [26] for 4× stereo SR. As shown in Figure. 1, it's evident that in comparison to SwinFSR [3], our LSSR achieves a state-of-the-art result with an impressive 89% reduction in parameters. And as shown in Table. 2, our LSSR use fewer Flops. This indicates that our LSSR is lightweight network.

We also provide the runtimes in Table. 2 (evaluated with 128 × 128 input on RTX 2080Ti GPU) to compare the computational complexity with SSRDE-FNet [6] and NAFSSR [5]. Our LSSR achieves

the best performance with a speedup of up to 3.07×. This highlights the fast and efficient nature of our LSSR.

**Table 2: The comparisons of lightweight and efficiency.**

| Models | PSNR | Flops | Tims(ms) | Speedup |
|---|---|---|---|---|
| SSRDE-FNet [6] | 26.43 | 65.89 G | 233.7 | 1.00x |
| NAFSSR [5] | 26.76 | 10.95G | 89.5 | 2.61x |
| **LSSR (Ours)** | 26.87 | 5.43G | 76.2 | 3.07x |

**Visual Comparison.** We show the visual comparisons for ×2 stereo SR on Flickr1024 [33], and ×4 on Middlebury and KITTI2012 in Figure. 6 7 8. These figures showcase that our LSSR effectively generates high-quality super-resolution images with intricate details and well-defined edges. In contrast, the other methods we compared may exhibit undesirable artifacts. This solidifies the proof of the effectiveness of our LSSR.

## 4.3 Ablation Study

**Lattice Stereo NAFBlock.** To confirm the effectiveness and versatility of the Lattice Stereo NAFBlock (LSNB), we conducted experiments on both the NAFSSR [5] model and the LSSR model, using LSNB and NAFBlock [4], respectively. Here, "w/o" denotes the usage of NAFBlock, while "w" signifies the utilization of LSNB. As shown in Table. 3, it is evident that the PSNR values for NAFSSR and LSSR without LSNB are 29.62 and 29.63, respectively. By replacing NAFBlock with LSNB, NAFSSR and LSSR achieve a PSNR improvement

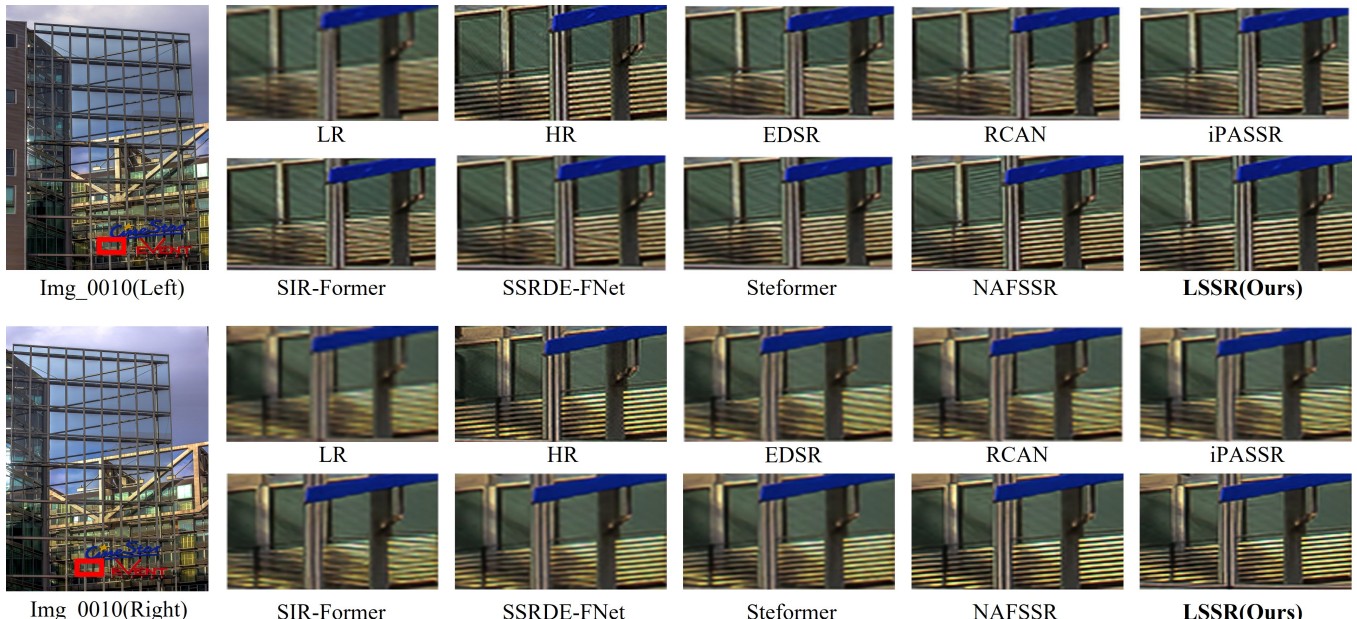

**Figure 6: Visual results (×2) achieved by different methods on the Flickr1024 dataset [33].**

**Table 3: The influence of lattice stereo NAFBlock (LSNB). We here report the results in PSNR for 4× SR. Noted that, "w/o" denotes the usage of NAFBlock, while "w" signifies the utilization of LSNB.**

| Modules | | PSNR | △ PSNR |
|---------|-----|-------|--------|
| NAFSSR | w/o | 29.62 | - |
| | w | 29.83 | +0.21 |
| LSSR | w/o | 29.63 | - |
| | w | 29.86 | +0.23 |

**Table 4: The influence of various RWBlock setting. We here report the results in both PSNR and SSIM for 2× SR.**

| Modules | GAP | STD | GAP+STD |
|---------|-----|-----|---------|
| PSNR | 35.08 | 35.12 | 35.39 |
| SSIM | 0.9413 | 0.9422 | 0.9530 |

of +0.21 dB, +0.23 dB respectively. This suggests that LSNB has the capacity to enhance the model's representation capabilities by adaptively learning the optimal combination patterns. Furthermore, it demonstrates that the LSNB can serve as a general concept applicable to other models, leading to performance improvements.

**Re-weight Block.** To learn the optimal combination pattern, we obtain re-weighted features using the attention-based mechanism RWBlock, which consists of two branches: global average pooling (GAP) and standard deviation (STD). As illustrated in Table. 4, the combination of both ensembles yields the best results.

**Table 5: The influence of different cross-attention modules. We here report the results in both PSNR and SSIM for 4× SR.**

| Modules | - | biPAM | SCAM | RCAM | **LSAM** |
|---------|------|-------|------|------|----------|
| PSNR | 23.41 | 23.63 | 23.76 | 23.75 | 23.87 |
| SSIM | 0.7192 | 0.7372 | 0.7419 | 0.7411 | 0.7432 |

**Lattice Stereo Attention Module.** To show the effectiveness of LSAM, we substitute the LSAM in LSSR with several SOTA approaches, including biPAM [34], SCAM [5], and RCAM [3]. As shown in Table. 5, compared with biPAM, SCAM, and RCAM, our LSAM achieves improvements of 0.24 dB, 0.11 dB, and 0.12 dB, respectively. When we examine LSSR without LSAM to assess the effect of the proposed LSAM on cross-view information, we find that our method achieves a 0.46 dB improvement with LSAM.

**Data augmentations.** We trained our model using different data augmentations to validate their effectiveness. As shown in Table. 6, introducing data augmentations such as random horizontal flip, random vertical flip, and channel shuffle has a positive impact on LSSR's performance. By employing all three data augmentations simultaneously, LSSR's PSNR increases from 23.45 dB to 23.87 dB, which is 0.11 dB better than using random flip alone.

**Single Input vs. Stereo Input.** StereoSR leverages supplementary data from cross-view images to significantly improve performance compared with SISR. To showcase the efficacy of stereo information in enhancing super-resolution performance, we conducted experiments using various input schemes. As indicated in Table 7, utilizing individual images during training results in a PSNR decrease of 0.45 dB compared to the baseline network. Likewise, when employing duplicated left images as inputs, the performance

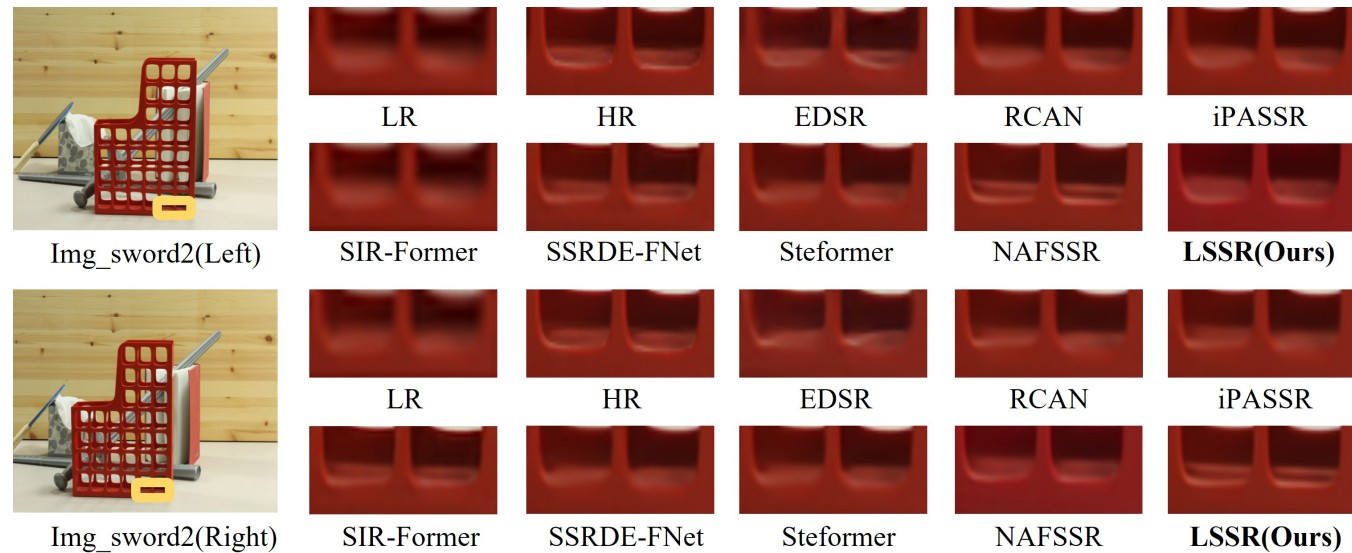

**Figure 7: Visual results of different methods for ×4 SR on the Middlebury dataset [28].**

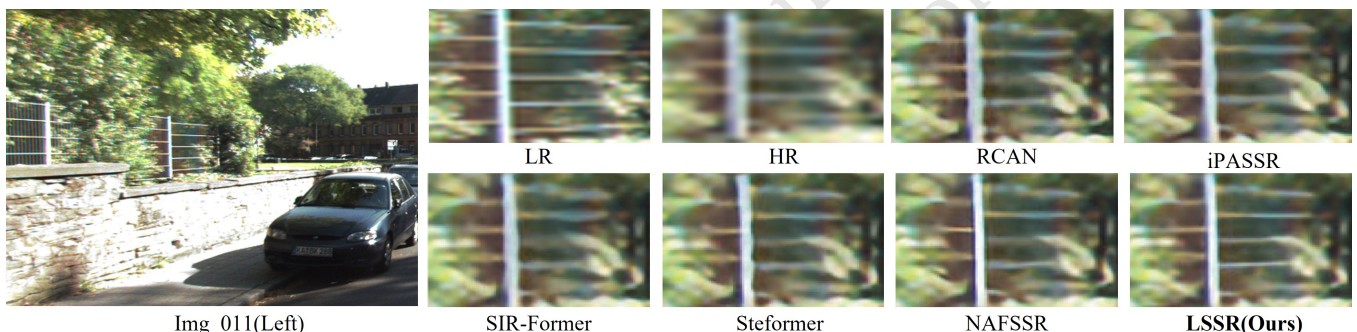

**Figure 8: Visual results of different methods for ×4 SR on the KITTI2012 dataset [8].**

**Table 6: 4× SR results (PSNR) with different data augmentations.**

| Horizontal flip | Vertical flip | Channel shuffle | PSNR | △PSNR |
|---|---|---|---|---|
| ✘ | ✘ | ✘ | 23.45 | - |
| ✔ | ✘ | ✘ | 23.66 | +0.21 |
| ✘ | ✔ | ✘ | 23.66 | +0.21 |
| ✘ | ✘ | ✔ | 23.65 | +0.20 |
| ✔ | ✔ | ✘ | 23.76 | +0.31 |
| ✔ | ✔ | ✔ | 23.87 | +0.42 |

**Table 7: Results achieved with different input schemes for 4× SR. Here, we report the results in both PSNR of the cropped left views.**

| Models | Inputs | PSNR |
|---|---|---|
| with single input | Left | 26.48 |
| with replicated inputs | Left-Left | 26.62 |
| LSSR | Left-Right | 26.93 |

of this modified configuration notably falls short of our initial network. These trials underscore the efficacy of our LSSR in capturing information from various perspectives.

## 5 CONCLUSION

In this paper, we propose a lattice structure that autonomously learn the optimal combination pattern of network blocks, avoiding the common practice of indiscriminately stacking numerous network blocks, ultimately presenting a lightweight model. Specifically, we design a lattice stereo NAFBlock (LSNB), which serves to bridge pairs of NAFBlocks by incorporating the re-weight block (RWBlock) through a coupled butterfly-style topological structure. RWBlock empowers LSNB with the capability to explore diverse combinations of pairwise NAFBlocks by utilizing the feature results obtained from RWBlock's adaptive re-weighting process. Furthermore, we propose a lattice stereo attention module (LSAM) facilitates the proper and accurate extraction of cross-view information. Extensive experimentation shows our LSSR surpass current models, establishing itself as a state-of-the-art performer in the field.

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
