# OpenReview forum: "Learning Optimal  Combination Patterns for Lightweight  Stereo Image Super-Resolution"
_acmmm.org/ACMMM/2024/Conference — MM2024 Poster_

### Official Review · Reviewer_B7JH · 2024-05-13

**Rating:** 3
**Confidence:** 3

**Summary:**

This paper studies the problem of how to effectively learn combination pattern instead of stacking massive network blocks in the context of stereo image super-resolution. Latticed adaptive reweighting is used to explore linear combination patterns between pairwise NAFBlocks, and lattice stereo attention module is employed to exchange information between stereo views. This design leads to lightweight network with good performance.

**Strengths:**

+ The idea of adaptive combination pattern learning and lightweight cross-view information exchange is interesting.
+ The authors have made every effort to introduce and explain the roles of LSNB and LSAM in their LSSR framework.
+ Promising experimental results are reported.

**Limitations:**

- Even though the network design looks reasonable, the paper lacks convincing evidence to describe what, how, and why diverse combination patterns can be learned and good information exchange can be achieved between stereo views to guide super-resolution.
- The experiments are not very convincing. Since the topic is about Stereo SR, an instance of clear improvement of the left image by some clue in the right image would be beneficial. But unfortunately such examples are missing. In addition, more details should be given to show why the Number of parameters is low in comparison to other methods (Fig. 1).
- Important ablation studies are missing. The parameter N (# of lattice layers) is closely related to the structure and performance of the network. The performance curve plot with respect to N would provide helpful understanding and judgement of the LSSR framework. However, even its value (N=?) is not given, let alone a definitely deserved ablation study.
- The presentation has some problems. In the “overall pipeline” paragraph, both low-resolution and high-resolution images have dimension HxWx3. In Figure 8, the hightlight block is missing.  In Table 4, only Middlebury results are given?

**Suitability:**

2

---

### Official Review · Reviewer_UEbb · 2024-05-21

**Rating:** 4
**Confidence:** 3

**Summary:**

This work proposes a Light Stereo Super-resolution Method (LSSR) that learns the optimal combination patterns of network blocks. This network is constructed based on the proposed LSNB and LSAM.

**Strengths:**

The comparison results show the LSSR model’s excellent performance.

The writing is clear and well-articulated, with detailed descriptions of LSNB, RWBlock, and LSAM.

**Limitations:**

1. As described in the Methods section, the LSNB can vary by adding or eliminating the NAFBlock. It is unclear how an LSNB block should select which variant to use.

2. The specific design features that help the proposed cross-attention module (LSAM) outperform other methods, as shown in Table 5, are not well explained. Is the whiten block a significant factor?

3. In Table 3, the comparison of parameters and FLOPs should be shown to ensure fairness in the experiments.

4. In Table 4, what would be the result if the GAP and STD were replaced with a simple MLP layer?

**Suitability:**

2

---

### Official Review · Reviewer_hPsL · 2024-05-23

**Rating:** 4
**Confidence:** 3

**Summary:**

The paper proposes a novel method for improving stereo image super-resolution (StereoSR) efficiency. It introduces a lattice structure to learn optimal combinations of network blocks, significantly reducing the computational load while maintaining competitive performance. The method leverages a new component, the lattice stereo NAFBlock (LSNB), which connects NAFBlocks using a re-weight block (RWBlock) in a butterfly-style topology to optimize the combination patterns dynamically.

**Strengths:**

The introduction of a lattice structure to autonomously learn the optimal combination of network blocks represents a significant shift from the traditional stacking of blocks, offering a fresh perspective on network design for StereoSR.

The proposed method uses butterfly-style topologies to enable efficient feature representation, which is theoretically sound and technically innovative. The lattice stereo attention module (LSAM) enhances cross-view feature integration, contributing to the model’s effectiveness.

Extensive experiments demonstrate that the LSSR model outperforms existing state-of-the-art methods with fewer parameters, providing robust evidence of its efficiency and effectiveness. The experiments cover various datasets and comparisons against multiple baseline models.

The paper is well-structured, with clear explanations of the novel components and their contributions to the overall system.

**Limitations:**

While the overall architecture is innovative, the individual components like NAFBlock and RWBlock are not new and have been used in similar contexts in previous works.

The method involves complex topological structures that might be challenging to implement and optimize in practice without significant expertise.

While the model is lightweight, the scalability of the approach when dealing with higher resolutions or significantly different stereo image configurations is not discussed.

**Suitability:**

2

---

### Official Review · Reviewer_k4wG · 2024-05-27

**Rating:** 2
**Confidence:** 3

**Summary:**

This paper introduces a novel stereo image super-resolution (stereoSR) method, aimed at enhancing the quality of super-resolution by leveraging auxiliary information from another viewpoint. The paper points out that existing stereo image super-resolution methods mainly focus on improving module design and stacking a large number of network blocks to extract and integrate information, but this approach leads to increased memory and computational costs. To address this issue, the paper proposes a lattice structure that can autonomously learn the optimal combination pattern of network blocks to achieve efficient and accurate feature representation, ultimately achieving the goal of lightweight stereo image super-resolution.

**Strengths:**

1.	By autonomously learning the optimal combination mode of network blocks, it avoids simply stacking a large number of network blocks, thereby reducing the number of parameters of the model and achieving lightweight.
2.	The proposed lattice structure design allows the network to process data more efficiently, reducing memory and computational costs, which is especially important for resource-constrained environments.
3.	Utilizing the Lattice Stereo NAFBlock (LSNB) and the Re-weight Block (RWBlock), the network can more accurately acquire and integrate feature representations, which is crucial for enhancing the quality of stereo image super-resolution.

**Limitations:**

1.Lacking innovation.The main architecture of this paper is very similar to NAFSSR, with a simple modification of the module. In the comparative experiments with NAFSSR, the performance does not show a significant improvement.
2. Please specify the version of NAFSSR used in the paper. The quantitative and qualitative results of NAFSSR (B) are superior to the proposed method.
3. In runtime comparison，the results of SSRDEFNet experiments in this paper differ significantly from those in the NAFSSR paper. Please explain the reason.

**Suitability:**

3

---

### Meta-Review · Area_Chair_Q3gf · 2024-07-04

**Recommendation:** Accept (Poster)
**Confidence:** 4

**Metareview:**

The paper proposes a new stereo image super-resolution method. The main contribution is a lattice structure that autonomously learns the optimal combination patterns of network blocks.
All reviewers agree that the experimental results show the proposed method outperforms SOTA with more efficient models.

The authors have addressed most of the concerns raised in the initial review.